# ArgRAG: Explainable Retrieval Augmented Generation using Quantitative Bipolar Argumentation

**Yuqicheng Zhu[1,2], Nico Potyka[3], Daniel Hernández[1], Yuan He[4], Zifeng Ding[5], Bo Xiong[6], Dongzhuoran Zhou[1,7], Evgeny Kharlamov[2,7], Steffen Staab[1,8]**

[1]*University of Stuttgart,* [2]*Bosch Center for AI,* [3]*Cardiff University,* [4]*University of Oxford,* [5]*University of Cambridge,* [6]*Stanford University,* [7]*University of Oslo,* [8]*University of Southampton*

**Editors:** Leilani H. Gilpin, Eleonora Giunchiglia, Pascal Hitzler, and Emile van Krieken

## Abstract

Retrieval-Augmented Generation (RAG) enhances large language models by incorporating external knowledge, yet suffers from critical limitations in high-stakes domains—namely, sensitivity to noisy or contradictory evidence and opaque, stochastic decision-making. We propose ArgRAG, an explainable, and contestable alternative that replaces black-box reasoning with structured inference using a Quantitative Bipolar Argumentation Framework (QBAF). ArgRAG constructs a QBAF from retrieved documents and performs deterministic reasoning under gradual semantics. This allows faithfully explanaining and contesting decisions. Evaluated on two fact verification benchmarks, PubHealth and RAGuard, Ar-gRAG achieves strong accuracy while significantly improving transparency.

## 1. Introduction

Retrieval-Augmented Generation (RAG) has emerged as a powerful framework that enhances the performance of large language models (LLMs) by incorporating external knowledge through a retrieve-then-generate paradigm (Khandelwal et al., 2019; Ram et al., 2023; Borgeaud et al., 2022; Izacard and Grave, 2021). By conditioning the generation process on retrieved documents, RAG systems can answer questions or complete tasks with broader and more up-to-date knowledge than what is stored in the model parameters alone (Siriwardhana et al., 2023; Lewis et al., 2020).

Despite its success, RAG faces several critical limitations that challenge its reliability and applicability in high-stakes scenarios. **First**, the retrieval process is imperfect. The underlying retriever is typically optimized for surface-level lexical or semantic similarity rather than for factual consistency or task-specific relevance (BehnamGhader et al., 2023). As a result, it may retrieve documents that are irrelevant, or contradictory—particularly in settings where misinformation and disinformation are prevalent (Deng et al., 2025; Khaliq et al., 2024). Such noisy retrieval can mislead the generator, as LLMs are highly sensitive to the input context and may incorporate spurious or conflicting information into their responses (Petroni et al., 2020; Cuconasu et al., 2024; Wan et al., 2024). **Second**, the decision-making process in RAG is inherently stochastic, driven by the probabilistic nature of autoregressive generation. This randomness can cause the model to produce incorrect or inconsistent outputs, even when relevant and accurate evidence is available (Longpre et al., 2021; Jiang et al., 2021; Potyka et al., 2024; He et al., 2025). Moreover, the reasoning behind the model's predictions is opaque: RAG systems may generate correct answers accompanied

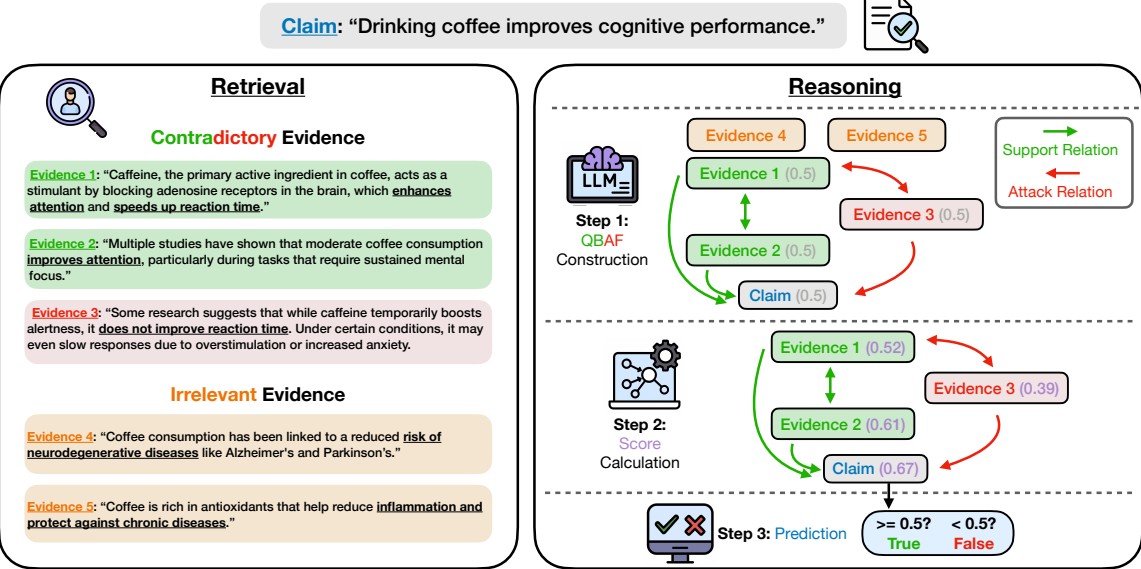

Figure 1: Overview of ArgRAG on a fact verification example. Given a claim, evidence (including supporting, contradictory, and irrelevant passages) is retrieved. In Step 1, ArgRAG assigns base scores to the claim and evidence, and identifies support and attack relations between them to construct a QBAF. In Step 2, final argument strengths are computed using QE gradual semantics over the QBAF. In Step 3, the claim is classified as true or false based on its final strength. The resulting QBAF provides a faithful explanation and supports contestability by allowing structured user interventions.

by explanations that may not faithfully reflect the decision process or retrieved evidence (Turpin et al., 2023; Stechly et al., 2025). These limitations raise serious concerns about the robustness, transparency, and reliability of RAG-based systems—especially in high-stakes domains such as healthcare, finance, or law. In such settings, accuracy alone is not sufficient; it is equally important for the system to be robust to noise, provide faithful explanations, and support trustworthy decision-making.

In this paper, we propose ArgRAG, an **explainable** and **contestable** alternative to standard RAG systems that rely solely on LLMs for reasoning. Our approach is based on a *Quantitative Bipolar Argumentation Framework* (QBAF) (Baroni et al., 2015); see Figure 1 for an overview. A QBAF consists of *arguments*, each assigned an *initial strength*, along with explicit *support* and *attack* relations between arguments. Rather than relying on the LLM to perform reasoning over retrieved documents, we use the LLM only to structure the retrieved documents into a QBAF. We then perform deterministic inference under *gradual semantics* to compute the final strengths of arguments. ArgRAG's reasoning process is inherently interpretable (Baroni et al., 2019) and facilitates faithful explanations via visualizations of the QBAF and the gradual reasoning process or via dialogues. Users can contest the decision made, for example by modifying assumptions about the inherent strength of arguments or

their relationship to other arguments as we illustrate in Section 4. In Section 5, we evaluate ArgRAG on two fact verification benchmarks–PubHealth and RAGuard, and show that it exhibits robustness to noisy and contradictory retrieved evidence.

## 2. Preliminaries

### 2.1. RAG-based Fact Verification

**Retrieval-Augmented Generation.** We let $\mathcal{T}$ denote the vocabulary of possible tokens. Given a sequence $t = (t_1, \ldots, t_m) \in \mathcal{T}^m$, *autoregressive language models* define a probability distribution over the sequence as:

$$P(t_1, \ldots, t_m) = \prod_{i=1}^{m} P_\theta(t_i \mid t_{<i}), \tag{1}$$

where $t_{<i} := (t_1, \ldots, t_{i-1})$ and $\theta$ are the model parameters. This formulation is typically implemented using the decoder part of Transformer models (Vaswani et al., 2017), as adopted by GPT models (Achiam et al., 2023).

RAG improves generation by retrieving relevant documents $(z_1, \ldots, z_k) \in \mathcal{Z}^k$ from an external corpus $\mathcal{Z}$ and conditioning generation on this context. Let $R_\mathcal{Z} : \mathcal{T}^* \to \mathcal{Z}^k$ be a retriever mapping a token sequence to $k$ retrieved documents. The RAG objective can be expressed as:

$$P(t_1, \ldots, t_m) = \prod_{i=1}^{m} P_\theta(t_i \mid t_{<i}, R_\mathcal{Z}(t_{<i})). \tag{2}$$

**RAG-based Fact Verification.** Given a natural language claim $c \in \mathcal{T}^*$, the objective is to determine whether it is true or false based on evidence retrieved from a large unstructured corpus $\mathcal{Z}$ (Bekoulis et al., 2021; Zhou et al., 2025a). In the common non-parametric setting (Ram et al., 2023), a retriever $R_\mathcal{Z}(c)$ returns a set of $k$ relevant documents $(z_1, \ldots, z_k)$, which are concatenated with the claim to form the input $[R_\mathcal{Z}(c); c]$ to the language model. The model then predicts a binary label $y \in \{\texttt{True}, \texttt{False}\}$ representing the veracity of the claim.

### 2.2. Quantitative Bipolar Argumentation Frameworks

QBAFs solve decision problems in an intuitive way by weighing up supporting and attacking arguments (Baroni et al., 2015).

**Definition 1 (QBAF)** *A* Quantitative Bipolar Argumentation Framework *is a quadruple $Q = (A, Att, Sup, \beta)$, where $A$ is a finite set of arguments, $Att \subseteq A \times A$ is a binary attack relation, and $Sup \subseteq A \times A$ is a binary support relation such that $Att \cap Sup = \emptyset$. The function $\beta : A \to [0,1]$ assigns a base score (apriori belief) to each argument.*

QBAFs compute a final strength $\sigma(a) \in [0,1]$ for each argument $a \in A$ under a specified *gradual semantics* (Rago et al., 2016; Amgoud and Ben-Naim, 2017; Baroni et al., 2019). These semantics are typically defined by an iterative update procedure that initializes the strength values with the base score and then iteratively updates the base scores based on the strength of attackers and supporters until they converge. Frequently applied semantics

include Df-QuAD (Rago et al., 2016), Euler-based (Amgoud and Ben-Naim, 2017) and quadratic energy (QE) (Potyka, 2018) semantics.

QBAF semantics can be compared based on semantical properties. The Euler-based semantics was motivated by the observation that strength values under Df-QuAD can saturate, which results in violation of some monotonicity properties. The Euler-based semantics avoided this issue, but introduced some new problems including an asymmetric treatment of attacks and supports. All these problems can be avoided by applying the QE semantics. However, more recently, it has been noted that Df-QuAD satisfies an interesting conservativeness property that neither the Euler-based nor the QE semantics satisfy (Potyka and Booth, 2024). The relevance of these properties depends on the application. We will focus on the QE semantics in the following since it satisfies almost all properties. We give a formal definition of the properties in Appendix A, and compare to Df-QuAD and Euler-based semantics in an ablation study later.

The QE update function updates arguments by aggregating the strength of attackers and supporters and using the aggregate to adapt the base score. The aggregate is computed as $E(a) = \sum_{b \in Sup(a)} \sigma(b) - \sum_{b \in Att(a)} \sigma(b)$ and updates the strength to $\beta(a) + (1 - \beta(a)) \cdot h(E(a)) - \beta(a) \cdot h(-E(a))$, where $h(x) = \frac{\max\{x,0\}^2}{1+\max\{x,0\}^2}$. The update process can be formulated as a *continuous dynamical system* to improve convergence of strength values (Potyka, 2018).

Intuitively, this formulation reflects a continuous tug-of-war between the argument's initial belief $\beta(a)$ and the influence of its context: supporters increase the score via $h(E(a))$, while attackers decrease it via $h(-E(a))$. As shown in Figure 2, the argument strengths evolve over time toward a stable equilibrium that balances these competing influences. Each line shows the strength trajectory of a specific argument from the example in Figure 1—the claim, two supporting evidence items (EV1, EV2), and one contradicting evidence item (EV3). All strengths are initialized uniformly at 0.5 and updated iteratively according to the QE

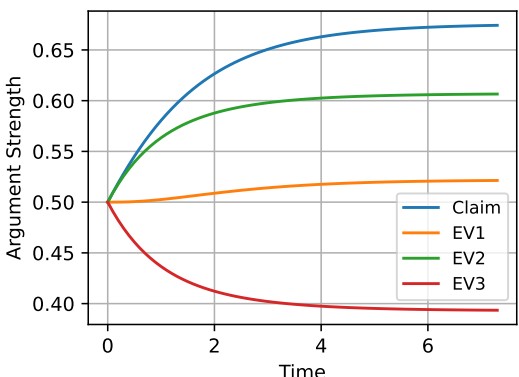

Figure 2: Evolution of argument strengths over time under QE semantics.

semantics. The system gradually converges, with the final strengths reflecting the overall argumentative structure encoded in the support and attack relations.

## 3. Argumentative Retrieval-Augmented Generation (ArgRAG)

This section introduces ArgRAG, detailing how we construct arguments, identify support and attack relations using LLM-based prompting, compute final argument strengths via QE semantics, and formalize properties that guarantee contestability. The overall procedure is summarized in Algorithm 1.

**Argument Construction.** In our method, we define the set of arguments as the claim and its top-$k$ retrieved evidence passages. Formally, given a claim $c$, we construct the

argument set $A = \{a_0, a_1, \ldots, a_k\}$, where $a_0$ corresponds to the claim itself, and $(a_1, \ldots, a_k)$ is the tuple of documents retrieved by a retriever $R_{\mathcal{Z}}(c)$. All arguments $a_i \in A$ are initialized with a uniform base score $\beta(a_i) = 0.5$, reflecting no prior bias.

**Relation Annotation** We use a two-step prompting procedure with an LLM to annotate the relations between arguments. The prompt templates used for each step are provided in Appendix B.2.

- **Step 1: Identifying claim-evidence relations.** We prompt the LLM to classify each evidence item $a_i \in \{a_1 \ldots, a_k\}$ as either `support`, `contradict`, or `irrelevant` with respect to the claim $a_0$. Arguments labeled `irrelevant` are removed from the argument set, resulting in an updated set $A' \subseteq A$. For the remaining items, we add directed edges $(a_i, a_0)$ to either the support relation $Sup$ or the attack relation $Att$, based on the classification.

- **Step 2: Identifying evidence-evidence relations.** We then prompt the LLM once with all remaining evidence arguments in $A' \setminus \{a_0\}$ to identify pairwise relationships. The model returns two sets, containing pairs of evidence items that support or contradict each other, respectively. For each pair $(a_i, a_j)$ in either set, we add bidirectional edges $(a_i, a_j)$ and $(a_j, a_i)$ to the corresponding relation $Sup$ or $Att$.

**Fact Verification with ArgRAG.** The constructed QBAF consists of the updated argument set $A'$, the support and attack relations $Sup$ and $Att$, and the base score function $\beta$. To compute the final strength scores $\sigma$, we apply the QE gradual semantics (see Section 2.2). The final prediction is determined by the strength of the claim node $\sigma(a_0)$ compared against a predefined threshold $\tau$ (default: 0.5). If $\sigma(a_0) \geq \tau$, the claim is classified as true; otherwise, it is false. In cases where all retrieved evidence is labeled as `irrelevant`, we fallback to directly querying the LLM without retrieved context.

## 4. Explanation and Contestation

ArgRAG is inherently explainable and contestable. While LLMs can generally be asked for explanations for their decisions, it remains unclear how faithful the explanations are. That is, the explanation may not be aligned with the reasoning process, and it may just rationalize the decision instead of explaining it faithfully. Since ArgRAG is based on explicitly creating an argumentation framework and reasoning about it, we can explain the decision faithfully.

**Explanation.** When the generated QBAF is of manageable size, users can directly interpret the decision based on the strength of pro and contra arguments by inspecting a visualization like in Figure 1. To gain deeper insight into how these strengths are determined, users can examine the strength evolution over time, as shown in Figure 2. For example, we can see how the strength of evidence 1 initially increases slower due to the fact that its attacker (evidence 3) and supporter (evidence 2) are similarly strong. As the process continues, evidence 3 is continuously weakened, while evidence 1 is strengthened, which results in a faster increase until the strength values eventually converge.

When the QBAF is too large or the user prefers not to examine low-level details, we can generate dialogue-based explanations similar to Cocarascu et al. (2019). To explain

---

**Algorithm 1:** ArgRAG for Fact Verification

---

**Input:** Claim $c$, Retriever $R_{\mathcal{Z}}$, Threshold $\tau$ (default: 0.5), a backbone LLM.
**Output:** Prediction $y \in \{\texttt{True}, \texttt{False}\}$, QBAF $Q = (A, Att, Sup, \beta)$, Final Argument
    Strengths $\sigma$.

**Argument Construction:**
Retrieve top-$k$ evidence documents: $(z_1, \ldots, z_k) \leftarrow R_{\mathcal{Z}}(c)$;
Construct argument set: $A = \{a_0, a_1, \ldots, a_k\}$ where $a_0 \leftarrow c$, $a_i \leftarrow z_i$ for $i = 1, \ldots, k$;
Initialize base strengths: $\beta(a) \leftarrow 0.5$ for all $a \in A$;

**Relation Annotation:**
**Step 1: Claim-Evidence Relations**
Use LLM to classify each $a_i \in \{a_1 \ldots, a_k\}$ as $\texttt{support}$, $\texttt{contradict}$, or $\texttt{irrelevant}$
 w.r.t. $a_0$;
Remove $\texttt{irrelevant}$ arguments from $A$ to get $A' \subseteq A$;
Add directed edges $(a_i, a_0)$ to $Sup$ or $Att$ according to classification of remaining $a_i$;
**Step 2: Evidence-Evidence Relations**
Prompt the LLM once with all evidence in $A' \setminus \{a_0\}$ to identify $\texttt{support}$ and
 $\texttt{contradict}$ pairs, returned as sets $SupPairs$ and $AttPairs$;
**foreach** $(a_i, a_j) \in SupPairs$ **do**
 |  Add bidirectional edges $(a_i, a_j)$ and $(a_j, a_i)$ to $Sup$;
**end**
**foreach** $(a_i, a_j) \in AttPairs$ **do**
 |  Add bidirectional edges $(a_i, a_j)$ and $(a_j, a_i)$ to $Att$;
**end**

**Strength Computation**
Apply QE gradual semantics to compute final strength $\sigma(a)$ for all $a \in A$;

**Prediction:**
**if** $A' \setminus \{a_0\} = \emptyset$ **then**
 |  $y \leftarrow$ LLM output for verifying $a_0$;
**else**
 |  $y \leftarrow \texttt{True}$ if $\sigma(a_0) \geq \tau$, else $\texttt{False}$;
**end**

**return** $y$, $Q$, $\sigma$

---

the strength of an argument $a$ with $\sigma(a) \geq 0.5$, we first identify its strongest supporter $Sup^*(a)$ and attacker $Att^*(a)$, and generate an explanation of the form: "*a is accepted because [Sup*(a)] even though [Att*(a)].*" If no attackers are present, the "even though" clause is omitted. For arguments with $\sigma(a) < 0.5$, we replace *accepted* with *rejected* and switch the roles of supporters and attackers. For example, in Figure 1, our explanation for accepting the claim is of the form "[Claim] is accepted because [Evidence 2] even though [Evidence 3]". For evidence 3, the explanation would be of the form "[Evidence 3] is rejected because [Evidence 2]".

**Contestation.** ArgRAG supports user intervention by enabling two natural forms of contestation:

- **Contest Base Score**: A user may disagree with a base score and ask the system to lower or to increase it.

- **Contest Polarity**: A user may disagree with the polarity of an argument. For example, an argument may have been classified falsely as an attacker, and the user may want to change its polarity to neutral or support.

In both cases, the changes can be incorporated into the QBAF automatically, and the result can be recomputed and presented to the user. For example, based on the quality of the studies associated with evidence 1 and 3, the user may want to change their base scores to 0.1 and 0.9. The strength of evidence 1, 2 and 3 will then change to 0.09, 0.5 and 0.89, and the strength of the claim will drop to 0.46. Hence, under these assumptions, the claim should not be accepted.

## 5. Experiments

### 5.1. Experimental Settings

**Datasets.** We evaluate our method on two fact verification datasets. The first is Pub-Health (Kotonya and Toni, 2020), a dataset for explainable fact-checking of public health claims. It contains 11,832 claims covering a wide range of topics, including biomedical science and government healthcare policy. Following the SELF-RAG (Asai et al., 2024) setup, we retrieve evidence for each claim from the English Wikipedia dump from Dec. 20, 2018 (Lee et al., 2019), preprocessed by Karpukhin et al. (2020) for information retrieval. The second dataset is RAGuard (Zeng et al., 2025), which targets fact verification in settings with misleading, contradictory, or irrelevant information. It includes 2,648 political claims made by U.S. presidential candidates from 2000 to 2024, each labeled as true or false. The associated evidence corpus contains 16,331 documents sourced from Reddit discussions, capturing naturally occurring misinformation and diverse viewpoints.

**Retrieval and Generation Pipeline.** Our RAG pipeline uses Contriever-MS MARCO (Izacard et al., 2022) as the default dense retriever. We retrieve the top 5 or top 10 documents per query to support generation. For text generation, we employ GPT-familiar LLM backbones (Achiam et al., 2023), including GPT-3.5 Turbo, GPT-4o-mini, and GPT-4.1-mini. These models are chosen for their strong instruction-following abilities, consistent API behavior, and a favorable trade-off between generation quality and computational efficiency.

**Baselines.** We compare our method against five baselines. The *w/o Retriever* baseline directly prompts the LLM without any external evidence. *IC-RALM* (Ram et al., 2023) follows the standard retrieval-augmented generation setup with in-context evidence. *IC-RALM+*, inspired by Adeyemi et al. (2024), augments this by explicitly prompting the LLM to identify and ignore irrelevant evidence before answering. *EXP* (Fontana et al., 2025) extends IC-RALM by asking the model to produce both a natural language explanation and a trustworthiness score (0–100) based on the retrieved evidence. Finally, CoT builds on EXP by incorporating Chain-of-Thought prompting (Wei et al., 2022) to encourage multi-step reasoning. Full prompt templates and configurations are provided in Appendix B.1.

| | PubHealth | | | RAGuard | | |
|---|---|---|---|---|---|---|
| | **GPT-3.5** | **GPT-4o-mini** | **GPT-4.1-mini** | **GPT-3.5** | **GPT-4o-mini** | **GPT-4.1-mini** |
| w/o Retriever | 0.826 | 0.840 | 0.838 | 0.768 | 0.771 | 0.725 |
| IC-RALM (Top5) | 0.700 | 0.671 | 0.654 | 0.694 | 0.697 | 0.685 |
| IC-RALM (Top10) | 0.735 | 0.722 | 0.670 | 0.697 | 0.713 | 0.716 |
| IC-RALM+ (Top5) | 0.734 | 0.673 | 0.637 | 0.691 | 0.700 | 0.694 |
| IC-RALM+ (Top10) | 0.756 | 0.711 | 0.654 | 0.691 | 0.713 | 0.703 |
| EXP (Top5) | 0.621 | 0.741 | 0.630 | 0.621 | 0.731 | 0.725 |
| EXP (Top10) | 0.617 | 0.754 | 0.648 | 0.627 | 0.752 | 0.737 |
| CoT (Top5) | 0.618 | 0.741 | 0.641 | 0.607 | 0.709 | 0.718 |
| CoT (Top10) | 0.618 | 0.749 | 0.647 | 0.618 | 0.755 | 0.719 |
| ARGRAG (Top5) | 0.835 | 0.846 | 0.892 | 0.801 | 0.805 | 0.803 |
| ARGRAG (Top10) | **0.838** | **0.855** | **0.898** | **0.804** | **0.813** | **0.804** |

Table 1: Overall comparison of all baseline and proposed methods on the PubHealth and RAGuard datasets, using three LLM backbones (GPT-3.5, GPT-4o-mini, GPT-4.1-mini). We evaluate each method with Top-5 and Top-10 retrieved documents. The best result in each setting is highlighted in **bold**, and the second best is underlined, and our method is shaded in gray for emphasis.

**Implementation Details.** Retrieval is implemented using LlamaIndex (Liu, 2022), which handles document chunking, preprocessing, and query routing. For fast nearest-neighbor search over dense embeddings, we integrate FAISS (Douze et al., 2024; Johnson et al., 2019) as the vector index. Language model inference is conducted via the OpenAI API (OpenAI, 2025) with temperature set to 0 for deterministic outputs. We use a maximum token limit of 15 for w/o Retriever, IC-RALM, and IC-RALM+; 4096 for EXP and CoT; and 2048 for generating QBAFs in ARGRAG. Our QBAF is implemented using the Uncertainpy library with quadratic energy gradual semantics. We use the RK4 algorithm for computing strength values, using step size $\delta = 0.1$ and termination condition $\epsilon = 0.001$.

## 5.2. Results and Analysis

**Overall Performance.** Table 1 reports the accuracy of all evaluated methods on the PubHealth and RAGuard datasets using three different GPT backbones. We first observe that all baseline RAG methods perform worse than the no-retrieval baseline, a finding consistent with prior work (Asai et al., 2024; Vladika et al., 2025). This highlights a known limitation of RAG systems in fact verification: when relying solely on LLM generation, the model becomes highly sensitive to noisy or contradictory evidence retrieved from external sources. In contrast, ARGRAG **consistently achieves the highest accuracy across both datasets and all LLM backbones**, demonstrating the robustness and effectiveness of structured argumentative reasoning within the RAG framework. Notably, ARGRAG significantly outperforms other RAG-based baseline methods and is the only RAG-based method that outperforms the no-retrieval baseline across all settings. Methods incorporating explanation and reasoning (EXP and CoT) tend to perform comparably or slightly better than standard RAG (IC-RALM, IC-RALM+) in most cases. This suggests that prompting the LLM to reflect via explanations or Chain-of-Thought reasoning can help mitigate the impact of noisy or contradictory evidence. However, their performance is in-

consistent; for example, both methods underperform standard RAG on PubHealth when using GPT-3.5. This variability and lack of interpretability underscore the need for more principled and transparent reasoning mechanisms, such as the one employed in ARGRAG.

Finally, we observe a modest performance gain when increasing the number of retrieved documents from Top-5 to Top-10, indicating that access to additional evidence can be beneficial—so long as the reasoning component is capable of handling the added complexity. ARGRAG maintains strong performance across both retrieval depths, further validating its robustness to varying evidence quantity and quality.

**Ablation Study.** Figure 3 presents an ablation study evaluating the impact of relation annotation strategies, base score initialization, gradual semantics, and prompt sensitivity. For all experiments, we use three semantically equivalent but syntactically varied prompts, generated by GPT-4o, to assess prompt sensitivity. We report the mean and standard deviation of results across these prompt variants in the figure.

First, we compare using only claim-evidence relations (EV2C) with the full argument graph including both claim-evidence and evidence-evidence relations (FULL). While EV2C already outperforms the no-retriever baseline, the FULL variant

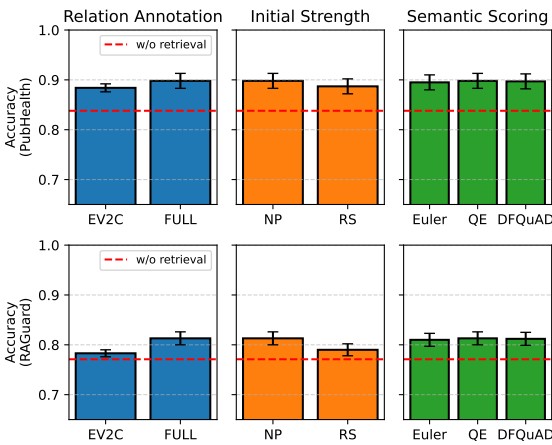

Figure 3: Ablation Study.

achieves consistently higher accuracy, particularly on RAGuard. This suggests that modeling interactions between evidence items is critical for handling conflicting evidence. Interestingly, EV2C exhibits lower variance under prompt changes, likely due to requiring fewer annotated relations. Second, initializing base scores without prior knowledge (NP) outperforms using retriever scores (RS), indicating that retriever confidence may not align with the actual relevance or trustworthiness of retrieved content. Finally, we observe only minor performance differences across different gradual semantics (Euler (Amgoud and Ben-Naim, 2017), QE (Potyka, 2018), DFQuAD(Rago et al., 2016)), suggesting that the choice of semantics can be guided by interpretability or computational considerations without sacrificing accuracy.

## 6. Related Work

Recent studies have shown that the performance of LLMs can degrade in the presence of irrelevant or contradictory context (Petroni et al., 2020; Cuconasu et al., 2024; Wan et al., 2024; Longpre et al., 2021; Zeng et al., 2025). To mitigate this, prior work has focused on fine-tuning either the retriever alone or both the retriever and the backbone LLM to improve retriever-generator alignment and re-rank retrieved documents by task relevance (Yoran et al., 2024; Wang et al., 2025; Ke et al., 2024; Asai et al., 2024). However, such fine-tuning requires high-quality training data and cannot be directly applied to proprietary models like the GPT series.

Other approaches aim to improve LLM reasoning by inducing intermediate graph-like structures (Wei et al., 2022; Yao et al., 2023; Besta et al., 2024; Marks et al., 2025; Zhou et al., 2025b). However, these methods do not guarantee a faithful alignment between the reasoning steps and the final output, as the entire process is embedded within the model's autoregressive decoding (Turpin et al., 2023; Xia et al., 2025; Stechly et al., 2025). In contrast, our method provides faithful and deterministic explanations by structuring the reasoning process explicitly through QBAFs.

The most closely related work is ArgLLM (Freedman et al., 2025), which also employs deterministic reasoning via QBAFs to enhance LLM decision-making. However, while our arguments are created from a retrieval engine, ArgLLM generates arguments from LLMs. To do so, ArgLLM asks an LLM for one argument that supports/attacks the claim if possible. The default QBAF will therefore contain at most three arguments. QBAFs of increasing depth are defined by recursively asking for a supporter/attacker of generated arguments. Another important difference to our work is that ArgLLM does not consider relationships between generated arguments, whereas we consider potential attacks and supports between arguments. The authors also prove some contestability guarantees, which, intuitively, state that adding a pro/contra argument or increasing its base score will make the claim stronger/weaker as desired. The guarantees hold for both Df-QuAD and QE semantics, but as shown in the appendix of Freedman et al. (2024), the QE semantics gives slightly stronger guarantees. However, experimentally, they found that both semantics work similarly well, which is also what we observed for ArgRAG in the RAG setting.

## 7. Discussion and Conclusion

In this paper, we introduced ArgRAG, a training-free neurosymbolic framework that combines RAG with QBAF for robust, explainable, and contestable fact verification. By modeling support and attack relations between the claim and retrieved evidence, and computing final argument strengths using QE semantics, ArgRAG achieves strong performance under noisy or conflicting retrieval. The resulting QBAF serves as a faithful explanation of the prediction and supports contestability, enabling human oversight and intervention.

In future work, we aim to improve ArgRAG in several directions. First, we currently treat the claim and each retrieved document chunk as a single argument; however, depending on the chunk length, a single retrieved passage may contain multiple, even contradictory arguments. In future work, we plan to apply techniques from argument mining (Lippi and Torroni, 2016) to extract finer-grained arguments within each document. Second, since using retriever scores for base initialization underperforms uniform initialization in our experiment, we will explore re-ranking methods (Zhuang et al., 2023; Kim and Lee, 2024) to assign more reliable base scores. Finally, while our current framework focuses on handling noise and contradictions in external knowledge, conflicts may also arise within the LLM's internal knowledge or between internal and external sources (Longpre et al., 2021; Xu et al., 2024). Incorporating internal knowledge extracted from the LLM into the QBAF—alongside external evidence—could enable argumentative reasoning over both sources and help resolve a broader range of knowledge conflicts in the RAG setting.

## Acknowledgments

The authors thank the International Max Planck Research School for Intelligent Systems (IMPRS-IS) for supporting Yuqicheng Zhu. The work was partially supported by EU Projects Graph Massivizer (GA 101093202), enRichMyData (GA 101070284) and SMARTY (GA 101140087), as well as the Deutsche Forschungsgemeinschaft (DFG, German Research Foundation) – SFB 1574 – 471687386. Zifeng Ding receives funding from the European Research Council (ERC) under the European Union's Horizon 2020 Research and Innovation programme grant AVeriTeC (Grant agreement No. 865958).

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

## Appendix A. Additional Properties of ArgRAG

In this section, we present key axiomatic properties satisfied by the QE gradual semantics used in ArgRAG.

**Property 3 (Neutrality)** *Let $Q = (A, Att, Sup, \beta)$ be a QBAF with final strengths $\sigma$. Suppose there exist $a, b \in A$ such that $Att(a) \subseteq Att(b)$, $Sup(a) \subseteq Sup(b)$, $Att(a) \cup Sup(a) = Att(b) \cup Sup(b) \cup \{d\}$ for some $d \in A$, and $\beta(d) = 0$, then $\sigma(a) = \sigma(b)$.*

This *Neutrality* property ensures that adding an attacker or supporter with no influence (i.e., zero base score) to an argument does not affect its final strength. For instance, if irrelevant evidence is not filtered out during Step 1 of relation annotation, a user can still mitigate its influence by manually assigning it a base score of zero.

**Property 4 (Monotony)** *Let $Q = (A, Att, Sup, \beta)$ be a QBAF with final strengths $\sigma$. Suppose there exist $a, b \in A$ such that $0 < \beta(a) = \beta(b) < 1$, $Att(a) \subseteq Att(b)$, $Sup(a) \supseteq Sup(b)$, then $\sigma(a) \geq \sigma(b)$.*

*Monotony* guarantees that, all else being equal, an argument with fewer attackers and more supporters will not lead to a lower final strength. In the context of ArgRAG, if a piece of evidence is updated (e.g., via reranking or user intervention) to gain more support or face fewer contradictions, its influence on the claim should appropriately increase or at least not decrease.

**Property 5 (Franklin)** *Let $Q = (A, Att, Sup, \beta)$ be a QBAF with final strengths $\sigma$. Suppose there exist $a, b \in A$ such that $\beta(a) = \beta(b)$, $Att(a) = Att(b) \cup \{x\}$, $Sup(a) = Sup(b) \cup \{y\}$ for some $x, y \in A$, and $\sigma(x) = \sigma(y)$, then $\sigma(a) = \sigma(b)$.*

The *Franklin* property ensures that if two arguments differ only by one additional attacker and one additional supporter, and both of these new arguments have equal strength, then the influence cancels out—leaving the final strength of the two arguments unchanged. In other words, equal and opposite evidence should balance out. Suppose two retrieved evidence passages each reference similar sources, but one supports the claim and another that contradicts it with equal strength. When both are added symmetrically to the QBAF, their effects should neutralize. This reassures users that adding balanced arguments won't arbitrarily shift the decision, which is especially useful for contestability.

**Property 6 (Weakening)** *Let $Q = (A, Att, Sup, \beta)$ be a QBAF with final strengths $\sigma$. Let $a \in A$ with $\beta(a) > 0$. Suppose that $f : Sup(a) \to Att(a)$ is an injective function such that $\sigma(x) \leq \sigma(f(x))$ for all $x \in Sup(a)$ and*

- *$Att^+(a) \setminus f(Sup(a)) \neq \emptyset$, where $Att^+ = \{b \in Att(a) \mid \sigma(b) > 0\}$;*

- *or there is an $x \in Sup(a)$ such that $\sigma(x) < \sigma(f(x))$,*

*then $\sigma(a) < \beta(a)$.*

**Property 7 (Strengthening)** *Let $Q = (A, Att, Sup, \beta)$ be a QBAF with final strengths $\sigma$. Let $a \in A$ with $\beta(a) < 1$. Suppose that $f : Att(a) \rightarrow Sup(a)$ is an injective function such that $\sigma(x) \leq \sigma(f(x))$ for all $x \in Att(a)$ and*

- *$Sup^+(a) \setminus f(Att(a)) \neq \emptyset$, where $Sup^+ = \{b \in Sup(a) \mid \sigma(b) > 0\}$;*

- *or there is an $x \in Att(a)$ such that $\sigma(x) < \sigma(f(x))$,*

*then $\sigma(a) > \beta(a)$.*

*Weakening* guarantees that if the strength of an argument's attackers dominates that of its supporters, its final strength decreases below its base score. Conversely, *Strengthening* ensures that when supporters outweigh attackers, the argument's final strength increases.

**Property 8 (Duality)** *Let $Q = (A, Att, Sup, \beta)$ be a QBAF with final strengths $\sigma$. Let $a, b \in A$ such that $\beta(a) = 0.5 + \epsilon$, $\beta(b) = 0.5 - \epsilon$ for some $\epsilon \in [0, 0.5]$. If there are bijections $f : Att(a) \rightarrow Sup(b)$, $g : Sup(a) \rightarrow Att(b)$ such that $\sigma(x) = \sigma(f(x))$ and $\sigma(y) = \sigma(g(y))$ for all $x \in Att(a), y \in Sup(a)$, then $\sigma(a) - \beta(a) = \sigma(b) - \beta(b)$.*

The *Duality* property ensures that symmetric argumentation structures with opposing priors lead to symmetrically shifted outcomes. Specifically, if two arguments $a$ and $b$ have base scores that are equally offset from 0.5 in opposite directions, and their sets of attackers and supporters are structurally mirrored with equal strengths, then their final strength deviations from the base scores are also mirrored.

## Appendix B. Prompt Templates

### B.1. Prompt Templates for Baseline Methods

```
You are a fact-checking expert.

For each input claim, output only true if the claim is factually correct,
or false if it is not.  Respond with a single word (true or false) | no
explanations, justifications, or additional text.

Claim:  {claim}
Answer:
```

```
You are a fact-checking expert.

Given a claim and retrieved evidence, output true if the claim is factually
supported, or false if it is not.  Base your answer only on the provided
evidence and your own knowledge if necessary.  Respond with a single word
(true or false) | no explanations, reasoning, or additional text.

Claim:  {claim}
Evidence:  {evidence}
Answer:
```

You are a fact-checking expert.

Given a claim and retrieved evidence, output true if the claim is factually
supported, or false if it is not.  Base your answer primarily on the provided
evidence, using your own knowledge only if necessary.  Consider only evidence
that is directly relevant to the claim.  Respond with a single word (true or
false) | no explanations, reasoning, or additional text.

Claim:  {claim}
Evidence:  {evidence}
Answer:

---

You are a fact-checking expert.

Your task is to evaluate the truthfulness of a claim based on the provided
evidence.  You must provide a score from 0 to 100, where 0 represents
definitively False and 100 represents clearly True.  Your score should reflect
your assessment of the claim's truthfulness in relation to the evidence.

Return a JSON object with two keys:  first, your analysis in the "explanation"
key, then a comma, finally a score with the "score" key.  The score should
match the analysis and your assessment of the claim's truthfulness.

Claim:  {claim}
Evidence:  {evidence}

Remember to always follow the json format:  format You must always PROVIDE
ONLY A SINGLE JSON evaluating the truthfulness of the claim in relation to
the evidence.  DO NOT include markdown formatting (such as triple backticks or
`json` tags) in the output.

OUTPUT_FORMAT = '{"explanation":  <explanation>, "score":  <score>}'

```
You are a fact-checking expert.

Your task is to evaluate the truthfulness of a claim based on the provided
evidence.  You must provide a score from 0 to 100, where 0 represents
definitively False and 100 represents clearly True.  Your score should reflect
your assessment of the claim's truthfulness in relation to the evidence.

Return a JSON object with two keys:  first, your analysis in the "explanation"
key, then a comma, finally a score with the "score" key.  The score should
match the analysis and your assessment of the claim's truthfulness.

Claim:  {claim}
Evidence:  {evidence}

Remember to always follow the json format:  format You must always PROVIDE
ONLY A SINGLE JSON evaluating the truthfulness of the claim in relation to
the evidence.  DO NOT include markdown formatting (such as triple backticks or
'json' tags) in the output.  Let's think step by step.

OUTPUT_FORMAT = '{"explanation":  <explanation>, "score":  <score>}'
```

## B.2. Prompt Templates for ArgRAG

The prompt we use for Step 1 in relation annotation:

```
Task:  Given a claim and multiple pieces of evidence, analyze the relationships
between evidence with respect to the claim.

Instructions:
- Support:  Two evidence items that reinforce each other regarding the claim.
- Contradict:  Two evidence items that conflict with each other regarding the
claim.

Output Format:
Return a single JSON object with two keys:  "support" and "contradict", each
mapping to a list of evidence pairs.

Example format:
{"support":  [["E1", "E2"], ["E1", "E3"]], "contradict":  [["E2", "E3"]]}

Claim:  {claim}
Evidence:  {evidence}

You must always PROVIDE ONLY A SINGLE JSON without any additional explanation
or commentary.  DO NOT include markdown formatting (such as triple backticks or
'json' tags) in the output.
```

The prompt we use for Step 2 in relation annotation:

```
Task:  Given a claim and multiple pieces of evidence, classify each evidence as
"support", "contradict", or "irrelevant" to the claim.  Classify each evidence
as either supporting, contradicting, or irrelevant to the claim.

Instructions:
- Support:  Evidence that backs the claim.
- Contradict:  Evidence that counters or limits the claim.
- Irrelevant:  Evidence unrelated to the claim.

Output Format:
Return a single JSON object with three keys:  "support", "contradict", and
"irrelevant", each mapping to a list of evidence items.

Example Format:
{"support":  ["E1"], "contradict":  ["E3", "E4"], "irrelevant":  ["E2", "E5"]}.

Claim:  {claim}
Evidence:  {evidence}

You must always PROVIDE ONLY A SINGLE JSON without any additional explanation
or commentary.  DO NOT include markdown formatting (such as triple backticks or
'json' tags) in the output.
```

## Appendix C.  Analysis of Score Distributions

In Figure 4, we compare scores from the EXP, CoT, and the final argument strength $\sigma(a_0)$ from ARGRAG, using both Top-5 and Top-10 retrieved documents.  The final strengths produced by ARGRAG show a clearer separation between positive (True) and negative (False) claims, reflecting more consistent calibration and alignment with the ground truth. In contrast, EXP and CoT scores are less discriminative, with considerable overlap between positive and negative distributions.

Currently, our argumentation graph includes only support and attack relations between arguments.  In future work, we plan to extend this by incorporating richer types of relations such as *caused by*, and explore more sophisticated forms of probabilistic reasoning on these extended graphs (Zhu et al., 2023, 2024).

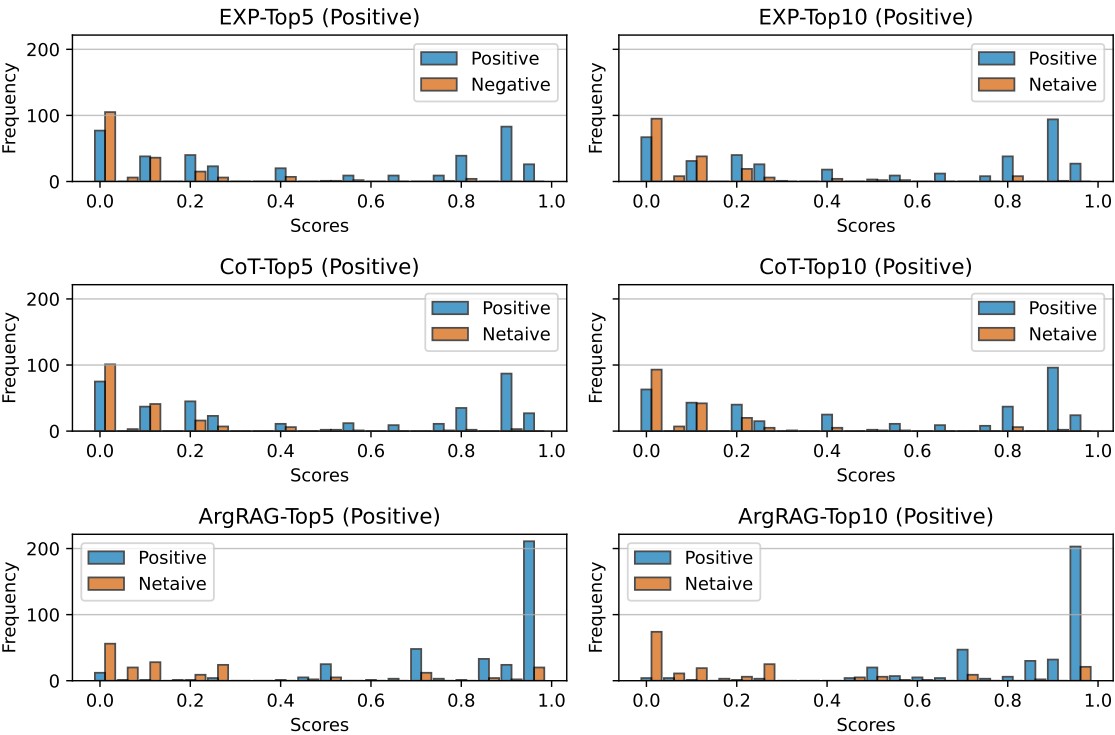

Figure 4: Distribution of scores from different methods on the PubHealth dataset using GPT-4.1-mini. We compare scores from the EXP, CoT, and the final argument strength $\sigma(a_0)$ from ARGRAG, using both Top-5 and Top-10 retrieved documents.

