# OpenReview forum: "ArgRAG: Explainable Retrieval Augmented Generation using Quantitative Bipolar Argumentation"
_nesyconf.org/NeSy/2025/Conference_Phase_2 — NeSy 2025 - Phase 2 Oral_

### Official Review · Reviewer_9gLT · 2025-07-01
**Good paper, but the claim about explainability should be rephrased**

**Rating:** 7
**Confidence:** 4

**Review:**

## Summary:
The paper proposes an approach based on Quantitative Bipolar Argumentation Frameworks (QBAFs) for Retrieval-Augmented Generation (RAG) with large language models.

## Strengths:
The paper is clearly written and well-structured. Its main contribution is clearly formulated and addresses a relevant challenge in the use of LLMs.

The proposed method is both novel and solid, tackling a significant problem in current RAG approaches. The experimental evaluation is fairly extensive, and comparisons with several baselines show that the proposed method outperforms state-of-the-art RAG techniques that do not incorporate reasoning.

## Weaknesses:
The only claim that seems somewhat overstated is the one regarding explainability and contestability. While it is clear that QBAFs can enable richer user interactions compared to traditional methods, this potential is only discussed theoretically and not supported by empirical evidence.

## Detailed comments:
The section on explainability and contestability lacks experimental validation and is not compared— even against simple baselines—using either explainability metrics or user studies. For example, the paper states:
"When the generated QBAF is of manageable size, users can directly interpret the decision based on the strength of pro and contra arguments by inspecting a visualization like in Figure 1."
However, what exactly qualifies as a "manageable size"? Specifically, what are the actual sizes of the QBAFs extracted from LLMs in your experiments? How often are they of a size that users would consider manageable?

The paper also mentions that when the QBAF is too large to be visualized directly, it can still be used to generate dialogue-based explanations by selecting only the strongest supporter or attacker. Why was this particular strategy chosen? Were alternative strategies explored?
Would the explanations remain equally convincing if the LLM were simply prompted to extract the strongest piece of evidence from the retrieved documents?

Without empirical investigations into these aspects, the claim regarding improved explainability remains purely theoretical and not particularly compelling. As such, it should be toned down or clearly marked as future work.

The claim regarding improved performance, on the other hand, is well-supported and convincing. I would recommend emphasizing this aspect and leaving a thorough evaluation of explainability and contestability to future work.

Minor Comments:
- At the end of Section 1, only Sections 4 and 5 are briefly described. Either provide a short overview of all the sections or remove that sentence altogether.
- Section 2 begins with three consecutive headings (Section 2 title, Subsection 2.1 title, and then a paragraph with its own heading). Consider inserting a short introductory text between these headings to improve the readability.

**Anonymity:**

Remain anonymous

---

### Official Review · Reviewer_XZx7 · 2025-07-09
**Well written and interesting paper on how to use argumentation to improve the faithfulness and interpretability of LLMs+RAG**

**Rating:** 8
**Confidence:** 4

**Review:**

Summary:

RAG has been often shown to improve the faithfulness of LLMs. However, it is far from being a perfect method. The limitations that the authors try to address in their paper are two: (i) the inherent imperfection of the retrieval process (i.e., wrong or simply irrelevant documents might be retrieved) and (ii) the lack of real reasoning performed by the LLM on these documents, which are processed by the LLM as any other input and then an output is generated auto regressively. To obviate to such problems the authors propose to use the LLM to generate a Quantitative Bipolar Argumentation Framework (QBAF) which is then used to reason over the retrieved documents. The approach is thus by definition more interpretable, as the user (in presence of small graphs) can actually inspect the QBAF and understand how the conclusion was drawn. The paper has been tested on two datasets PubHealth and RAGuard, and has been shown to improve the performance of different GPT-mini versions equipped with different retrievers.

Strengths:

1. The paper is very well written and easy to follow
2. The problem is of high relevance for the community
3. The approach is intuitive
4. The experimental analysis shows an improvement in performance
5. An ablation study has been conducted to study the impact of relation annotation strategies, base score initialization, gradual semantics, and prompt sensitivity.

Weaknesses:

1. At the beginning the authors try to explain why they picked a semantics over the others. The discussion is a bit difficult to follow without knowing the properties of the different semantics. Maybe the authors can simply state what are the nice properties of the QBAF and then do a better comparison in the Appendix.

2. The approach has only been tested with 5 and 10 documents. However, I am not sure about the scalability of the approach. What happens with 20 or 30 documents?

Questions:

1. What is the overhead introduced by the method in the experiments conducted? How does it compare with the other methods?

2. What is the time complexity of the approach?

**Anonymity:**

Remain anonymous

---

### Official Review · Reviewer_ADti · 2025-07-10
**The of argument structure analysis for claim verification is of high relevance. The proposed method shows some improvements (though not clear if statistically significant). The authors frame the proposed method as a RAG based method, however, it is not the LLM that generates the judgement about the claim but a QBAF algorithm. Baselines and additional experiments would make the paper stronger supporting the method's robustness w.r.t. evidence noise.**

**Rating:** 5
**Confidence:** 3

**Review:**

This paper tackles the task of claim verification. It proposes a solution based on retrieved evidence, a LLM, and the Quantitative Bipolar Argumentation Framework (QBAF). The LLM is used to build an argument graph among the claim and retrieved evidence. Then doing inference via gradual semantics over the graph a final decision True/False for the claim is derived.

Strength

- It tackles the challenging task of argument analysis where current LLMs struggle with analysing / reasoning over the complex interactions among the available pieces of evidence and the claim.

Weaknesses

- It is not clear whether it is appropriate to call the proposed method RAG-base method. Given that LLMs are used to assess textual relationships (contradict/attack/support/irrelevant) and the final prediction is based on the outcome of the QBAF algorithm not generated by an LLM.

- The authors motivate the approach with the issues that current models face when confronted with misleading, irrelevant and noisy evidence. However, neither the method nor the experimental setup seem to explicitly address this.

Other comments:

- Wikipedia as corpus for PubHealth?
- From results in Table 1, it seems not possible to support the statement made "This suggests that prompting the LLM to reflect via explanations or Chain-of-Thought reasoning can help mitigate the impact of noisy or contradictory evidence."
- Are the scores reported in Table 1 statistically significant? Differences are rather small. Also, on the variant from top-5 to top10 there seems to be negligible (no difference). This could be maybe a good thing? Because increasing to 10 could mean just adding noise or misleading information... then if the method maintains the same score means that is robust. Actually a study on the behaviour of the method w.r.t. to the proportion of supportive, contradictory, noise/misleading (these could be synthetically inserted) evidence passages would be useful to highlight the robustness of the methods.
- Ablation Study. Could not you initialise the claim node with 0.5, but the claim nodes with the strength of the labels supportive/contradict/irrelevant w.r.t. to the claim?
- Why there is no comparison in Table 1 with graph method discussed in related work, e.g., (Marks et al., 2025; Besta 2024; Freedman et al. 20204, etc.)?

**Anonymity:**

Remain anonymous

---

### Official Review · Reviewer_4P3a · 2025-07-11
**A well written paper with a promising approach that lacks certain evaluation dimensions**

**Rating:** 6
**Confidence:** 3

**Review:**

The paper presents a new variation of a neuro-symbolic hybrid pipeline in which, as a first step, an LLM does some premise instantiation/parsing/other textual manipulation and extracts structured representations/logic formulas/Prolog programs etc.; and then a deterministic reasoner/SAT solver/ASP system arrives at the final result.
This time the hybrid pipeline is instantiated as an LLM+search step that performs RAG, retrieving top relevant passages and assigning them support scores, and a QBAF reasoner.

The system preserves the usual benefits of hybrid systems: it is interpretable, transparent, verifiable, and a user may even intervene at the reasoning step.
The argumentation framework as the formal component of a hybrid system is especially compatible with fact verification as the evaluation task, which is why the proposed system, ArgRAG, is evaluated on two fact checking datasets.

ArgRAG is compared to 5 other systems including representative baselines for the system: a black-box LLM call, a simple RAG call, a RAG call that further filters out irrelevant passages etc. These "structural" baselines act as a sort of an ablation study, highlighting the value of the argumentation component.
Indeed, "ArgRAG consistently achieves the highest accuracy across both datasets and all LLM backbones". Although the difference with a native black-box LLM calls is at times marginal.

Both the system and the experiments are clearly described, and the results seem promising. ArgRAG outperforms other RAG-based baselines and is the only RAG method that outperforms the black-box LLM call. However, I missed comparing ArgRAG to any SOTA fact checking systems, and first and foremost, with ArgLLM, or with other hybrid systems that incorporate any sort of formal reasoning and not only LLM-based steps.
More so, other fact verification works are completely omitted from the related work section.

The lack of evaluation that positions the work in comparison to SOTA hybrid systems and SOTA fact checking systems is the main serious issue with the paper.

In addition, another discussion point I would prefer to be in the paper is how generalizable is the system architecture (and specifically the argumentation framework) to other application tasks beyond fact checking.

If these two issues are addressed, I believe this paper could be a nice addition to the conference. As it stands now, it is borderline.


Minor:
- page 2: "are computed using QE gradual semantics" --> the abbreviation QE has not yet been introduced

**Anonymity:**

Remain anonymous